# Effect of Gum Acacia on the Intestinal Bioavailability of n-3 Polyunsaturated Fatty Acids in Rats

**DOI:** 10.3390/biom12070975

**Published:** 2022-07-12

**Authors:** Leslie Couëdelo, Cécile Joseph, Hélène Abrous, Ikram Chamekh-Coelho, Carole Vaysse, Aurore Baury, Damien Guillemet

**Affiliations:** 1ITERG, Nutrition Health & Lipid Biochemistry Team, 33610 Bordeaux, France; h.abrous@iterg.com (H.A.); i.chamekh@iterg.com (I.C.-C.); c.vaysse@iterg.com (C.V.); 2ITERG, Formulation Team, 33610 Bordeaux, France; c.joseph@iterg.com; 3NEXIRA, CEDEX 3, 76723 Rouen, France; a.baury@nexira.com

**Keywords:** emulsion, intestinal lipid absorption, lymphatic lipids, fish oil, omega-3 bioavailability, PUFA absorption, acacia gum, acacia fiber, gut absorption fiber inhibitory effect, DHA

## Abstract

Lipid emulsification is a technique that is being explored for improving the bioavailability of omega 3 (n-3) long chain (LC) fatty acid (FA). The nature of the emulsifiers can differently impact the lipid bioavailability via a modification of the lipolysis step. Among natural emulsifiers, gum acacia (GA), an indigestible polysaccharide, provides protective encapsulation of n-3 by forming a specifically crown-like shape around lipid drops, which could also impact the digestion step. Despite the interest in lipolysis rate, the impact of GA on lipid bioavailability has never been explored in a complete physiological context. Thus, we followed in a kinetics study the n-3 bioavailability in rat lymph, orally administered DHA-rich oil, formulated based on GA compared to the bulk phase form of the oil. The AUC values were significantly improved by +121% for total TG and by 321% for n-3 PUFA, specifically for EPA (+244%) and for DHA (+345%). Benefits of GA have also been related to the transport of FA in lymph, which was 2 h earlier (Tmax = 4 h), compared to the Tmax (6 h) obtained with the bulk phase oil. All the data showed that GA is one of the most favorable candidates of natural emulsifiers to improve n-3 bioavailability and their rate of absorption for health targets.

## 1. Introduction

Improving the bioavailability of n-3 PUFA is of particular interest to human health, since according to the latest epidemiological study [1], the consumption of long chain (LC) n-3 PUFA is almost twice lower than expected from the French guidelines [2,3]. Yet, eicosapentaenoic acid (EPA, 20:5 n-3) and docosahexaenoic acid (DHA, 22:6 n-3) are recognized to reduce the risk of developing several chronic pathologies, such as cardiovascular, stroke, neurodegenerative, inflammatory and cancer diseases [4,5,6,7,8,9,10]. In this context, in addition to increasing dietary intake, new research is focusing on improving the bioavailability of n-3 LC PUFA.

It is well known that lipid bioavailability mainly depends on several factors inherent to the food matrix, including intramolecular and supramolecular structures [11,12,13,14,15]. However, recently, the process of lipid emulsification has been described to improve the intestinal uptake of oils and lipophilic compounds [11,12,13,14,15], as well as the bioavailability of n-3 PUFA in lymph or in plasma [12,16,17,18,19,20,21,22]. A lipid emulsion consists of a dispersion of small lipid droplets in an immiscible phase, often aqueous phase. Studies have showed that the formation of lipid droplets has a positive effect on the fatty acid (FA) bioavailability via improved gastrointestinal lipolysis. Indeed, lipid emulsification provides a large lipid–water interface for the adsorption of pancreatic lipase and promotes faster and more efficient lipolysis compared to the bulk phase oil [23,24,25]. The rate of lipolysis is, thus, strongly influenced by the supramolecular structures of the lipid matrix [15,26,27,28], making gastrointestinal lipolysis a crucial step for FA bioavailability. The improvement of intestinal absorption of fatty acids would result in a modification of lipid micellization and would favor the FA absorption within enterocytes and would promote the accretion of triglycerides (TG) into chylomicrons (CM) in rat lymph.

The composition of the interface, i.e., the nature of the emulsifiers, can affect the lymphatic metabolism of TG, and consequently the plasma concentrations of TG.

In general, emulsions are thermodynamically unstable and need interfacial agents to avoid the emulsion break. These agents provide consistency (thickening and gelling) and/or to stabilize the system or even for the flavour-controlled release. Several emulsifiers are used in food emulsions or in nutraceutical delivery systems. They can be adsorbed at oil–water interfaces to protect lipid droplets against coalescence. Due to their different interfacial properties, emulsifiers differently affect the enzymatic activity and lipolysis rate during lipid digestion [29], which results in a modification of FA bioavailability [16,24,25,30,31,32,33,34,35,36,37]. For example, casein and Tween have been described to inhibit, whereas lecithin would enhance lipid digestion [12,17].

Among food-grade surfactants, gum acacia (GA), a hydrocolloid exudate from acacia trees, is a polysaccharide carbonate and protein complex, which has been used for decades in food formulations for its interfacial properties. Hydrocolloids are water-soluble polymers with a wide variety of molecules, including proteins and polysaccharides. This natural product is a complex mixture of biopolymers that are heterogeneous of arabinogalactan polysaccharide with a wide range of molecular weight (Mw; from 0.02 to 11 × 10^6^ g.mol−1) [38,39]. The major class of compounds including arabinogalactan proteins (AGP), and glycoproteins (GP), are adsorbed at the oil–water interface, and thanks to their amphiphilic behaviors, they have emulsifying properties and are responsible for the oil stabilization processes [39,40,41].

The GA ability to form a coating layer surrounding the spherical submicrometric structure of lipids [42] confers to GA some encapsulation properties and protects labile lipids from evaporation and oxidation. Thus, GA has been extensively studied in formulation improvement of sensitive compounds and lipids, even for n-3 PUFA, with significant increase in oxidative stability in its final forms [43,44,45,46]. As the oxidative phenomena usually occurs at the oil–water interface (O/W), it has been observed that emulsifiers located at the O/W interface were more conducive to exert an antioxidant effect compared to their internal location in the lipid droplets [47]. GA contains 90% indigestible fiber, which confers GA some prebiotic activities in the colon. Notably, the microbiota taxonomy was modulated by increasing the *Lactobacillus* and *Bifidobacterium* populations and the synthesis of metabolites, the so called short chain fatty acids (CCFA) [48,49,50]. The specific structure of GA induced noticeable progressive colic fermentation, mainly at the distal level, which is responsible for the potent digestive tolerability [50,51].

In addition to the use of GA in industrial applications, such as emulsification and encapsulation, the structure complexity due to the formation of a solid crown-like shape surrounding the lipid droplets induces some controversial expectations about the putative impact of GA on the physiological bioavailability of n-3 PUFA. Firstly, the emulsification properties of GA could increase the intestinal uptake of n-3 LC-PUFA with regard to the greater interfacial area exposure to the digestion process. Indeed, in vitro studies have demonstrated a positive GA impact in comparative models on lipid bioavailability. However, in the in vitro studies, highly simplified conditions of free fatty acids released under the action of lipase and the simplified transit of these molecules through the mimicked intestinal barrier limit the assessment parameters [29,52].

Secondly, the indigestible and resistant crown-like shape of GA formed around the lipid droplets of n-3 PUFA latter in the gut transit delay or even could limit their uptake. In fact, some fiber, even if it is soluble, has been described to reduce body fat mass because of its affinity to bind to fats and to inhibit intestinal digestion [53,54].

Even if these data suggest that GA could be an interesting emulsifier for improving the oil bioavailability in comparison with other surfactants, some in vivo demonstrations involving a complete function of physiological digestion are still required.

Natural surfactants are steadily increasing in formulation. Thus, within the framework of the development of healthy food products based on natural n-3 PUFA (DHA) from microalgae, we sought to determine the influence of the FA encapsulation with a natural emulsifier GA on the intestinal absorption and bioavailability of n-3 LC-PUFA. We hypothesized that GA could improve the bioavailability of n-3 PUFA from algae oil, due to its specificity to form a crown-like shape around the lipid droplets.

In order to investigate the interest and the potential nutritional value of combining GA and algae oil, rats with a lymphatic duct shunt were fed with algae oil, supplied either in a bulk phase or in O/W emulsion that was GA-stabilized. Lymph was collected sequentially over 6-h post-feeding. Kinetic and lymph samples were characterized according to their fatty acid composition and quantification. For more precision, the dispersion state and stability of the emulsion have been determined to support the in vivo data.

## 2. Materials and Methods

### 2.1. Material

Microalgae oil Omegavie DHA 400 Algae QualitySilver^®^ was obtained from *Schizochytrium* sp. strain, provided by Polaris (Quimper, France). The lipidic characterization as fatty acid profile (Table 1), oil structure and glyceridic composition (Table 2) was determined according to the standardization methods of IUPAC 6.002 and NF EN 14105 and NF EN ISO 12966, respectively. The oil provides mainly DHA (47% of total GA) and to a lesser extent, EPA (1.5% of total GA) in the TG form (97%).

Inavea™ Pure Acacia Original was provided by Nexira (Rouen, France).

Acetic, formic, sulfuric and hydrochloric acids, sodium chloride (NaCl), potassium chloride (KCl), and sodium carbonate (Na_2_CO_3_) were provided by Thermo Fisher Scientific (Strasbourg, France). They also supplied the organic solvents used (analytical or HPLC grades). As internal standards, 1,2-diheptadecanoyl-sn-glycero-3-phosphatidylcholine (PC 17:0), 1,2,3-triheptadecanoyl-sn-glycerol (TG 17:0), and heptadecanoic acid (FFA 17:0) were obtained from Avanti Polar Lipids INC (Alabaster, AL, USA).

Ketamine, xylazine, buprenorphine and sodium pentobarbital were provided by Axience (Pantin, France) and lidocaine was supplied by Ceva (Libourne, France). Pentobarbital sodique (Exagon) and lidocaine (Xylovet) were provided by Ceva (Libourne, France).

Acetyl chloride, acetic acid, sodium chloride (NaCl), potassium chloride (KCl), and sodium carbonate (Na_2_CO_3_) were provided by Thermo Fisher Scientific (Strasbourg, France). They also supplied the organic solvents used, including acetonitrile, di-ethyl ether, ethanol, heptane, hexane and methanol (analytical or HPLC grades). As the internal standard, 1,2,3-triheptadecanoyl-sn-glycerol (TG 17:0) was obtained from Avanti Polar Lipids INC (Alabaster, AL, USA).

### 2.2. Lipid Formulations

The algae oil was encapsulated by NEXIRA (Rouen, France) with gum acacia. Optimization in term of process parameters was developed in accordance with the following three main observations: droplet diameter, stability of the emulsion (in liquid form) and oxidative status kinetics, as described below. Briefly, the oil phase was dispersed into aqueous phase containing preliminary dissolved GA, using a shearing device (Turrax-T50, 10.000 rpm; 3 min/L). Then, the coarse O/W emulsion was homogenized (Gaulin-LAB 60–10 TBS, 1 pass, 2 stages 240/40 bars).

Final composition of the encapsulated oil thanks to the emulsion was 10, 20 and 70 gr/kg, respectively, for algae oil, GA and water.

A direct visualization of the oil droplets was carried out extemporaneously and 5 days after the emulsification process by using an optical microscope (Axiostar^®^ with a water immersion × 100 objective; Zeiss, Germany) connected to an Axiocam 208 color digital video camera and controlled by the Zen Core software for image retrieval. The maximal droplet size represented 3µm after oil emulsification and did not vary for at least 5 days (Figure 1).

The lipid oxidation was determined 3 days after emulsification with regard to the peroxide and P-*anisidine* values (Table 3), determined by the French standardization method (NF EN ISO 3960 and NF EN ISO 660, European claim 2568/91 and COI/T20/Doc N°34/Rev.1 2017, respectively) and did not show any particular oxidation state.

### 2.3. Experimental Design: Animal and Surgical Procedures

Male Wistar rats (8 weeks-old, body weight 300–350 g) were obtained from Elevage Janvier (Saint-Berthevin, France) and were randomly assigned to one of the two dietary groups. Animals were treated in accordance with the European Communities Council Guidelines for the Care and Use of Laboratory Animals (2010/63/EU). All experiments conformed to the Guidelines for the Handling and Training of Laboratory Animals. The experiments and procedures were approved by the French ministry, recorded under the APAFIS n° 2017031014448864, and were carried out in compliance with the local ethics committee in Bordeaux, France (CEEA50).

Rats were housed for at least 3 days before the experiment in a controlled environment, with constant temperature and humidity, and with free access to food and water. The day before surgery, rats were fed a fat-free diet (Epinay, France) and had free access to water. On the day of surgery, each rat was placed under anesthesia by an injection of ketamine/xylazine (100/10 mg/kg, respectively). The abdomen was incised transversely on the left side. The organs were isolated to allow visualization of the main mesenteric lymph duct. A polyethylene catheter (0.95 mm × 15 cm Biotrol, Paris, France) was inserted into the lymphatic duct and secured by two ligatures, as described by Bollman et al. [55] and Couëdelo et al. [13]. The abdomen was then sutured and the rats were placed in individual restraining cages, in a warm environment with free access to water. To prevent pain, rats received an intra-peritoneal injection of buprenorphine (0.02 mg/kg) 1 h before and 2 h after surgery.

In each experiment, an equivalent amount of 12 mg and 380 mg of EPA and DHA, respectively, from microalgae oil either in bulk phase or emulsified with GA) were administered to rats (*n* = 8 rats per group) by one-shot oral gavage. Lymph was collected hourly for 6 h post feeding. At the end of the experiment, rats were euthanized by an intra-peritoneal injection of sodium pentobarbital and lidocaine.

The sequential collection of lymph allows us to define the kinetics of intestinal absorption of n-3, according to their formulation (in bulk phase vs. emulsified with GA), as well as the maximum lymphatic concentration (Cmax) and the time (Tmax), for which this maximal concentration of n-3 PUFA has been reached.

### 2.4. Fatty Acid Profile and N-3 LC PUFA Composition in Lymph

Total fatty acid composition from lymph was directly obtained by the method described by Lepage and Roy [56].

The resulting FA methyl esters (FAME) were analyzed by GC (TRACE GC, Thermo Scientific, Waltham, MA, USA), equipped with a flame ionization detector (FID) and a split injector. A fused-silica capillary column (BPX 70, 60 m × 0.25 mm i.d., 0.25 μm film; SGE, France) was used with hydrogen as a carrier gas (inlet pressure: 120 kPa). The split ratio was 1:33. The column temperature program was as follows: from 160 °C, the temperature increased to 180 °C at 1.3 °C/min, and was maintained for 65 min before increasing at 25 °C/min until 230 °C for 15 min. The injector and detector were maintained at 250 °C and 280 °C, respectively. GC peaks were integrated using Chromquest software (Thermofinnigan, Courtaboeuf, France). FA were quantified using tri-heptadecaenoic acid as an internal standard, and were added at 10% of the lipid weight before the (trans) methylation procedure. As palmitic and oleic acids are major endogenous FA in the lymph, only n-3 PUFA and myristic acid that were absent from the endogenous FA in lymph have been studied during the postprandial kinetics of FA absorption.

### 2.5. Statistical Analysis

Data were expressed as mean values with their standard deviation (SD) and were analyzed by XLStat software to evaluate the kinetics of absorption of n-3 LC-PUFAs, EPA and DHA over the 6 h period following lipid administration. Moreover, it has been determined through the kinetics curve (i) the area under the curve (AUC; expressed in mg × h/mL) to assess the amount of n-3 LC PUFAs (EPA and DHA) absorbed and (ii) the maximum lymphatic concentration of n-3 C-LC PUFAs (Cmax) and the maximum time (Tmax) to reach it. Data were analyzed between the two groups on two unpaired points, showing equality of variances (Shapiro–Wilk normality test). Intergroup comparisons were made on the basis of their respective mean by the parametric Student’s *t*-test; *p*-values lower than 0.05 were considered to be statistically significant.

## 3. Results

### 3.1. Influence of Using Gum Acacia as Emulsifier on the Intestinal Absorption of N-3 LC-PUFA

#### 3.1.1. Total Fatty Acid Absorption

The amount of total fatty acids was determined over the time-kinetics from 1 to 6 h after lipid administration, supplied by the algae oil either in bulk phase or emulsified with GA (Figure 2); the AUC, Tmax and Cmax values were reported in Table 4.

When lipids are supplied by the oil in bulk phase, the lipid concentration in lymph increased up to 6 h (Tmax) and reached a peak of FA absorption of 153 mg/mL lymph/g lipid intake (Cmax). On the other hand, when the oil is emulsified with GA, the absorption kinetics of fatty acids increased up to 4 h (Tmax) for a Cmax of 255 mg/mL lymph/g of lipid intake. After the Tmax, the FA absorption decreased in lymph up to 6 h.

Considering the AUC data (Figure 2), the data demonstrated that the lipid emulsification with GA (821 mg/mL/g of lipid intake.h) significantly improved (+120%; *p* < 0.05) the bioavailability of FA in the lymph compartment compared to the oil provided in bulk phase (375 mg/mL/g of lipid intake.h).

#### 3.1.2. Lymphatic Recovery of N-3 PUFA

Due to the high content of n-3 in microalgae oil (43% DHA), the bioavailability of n-3 PUFA, EPA and DHA was monitored in the lymph compartment.

Total n-3 PUFA absorption

The kinetics of n-3 PUFA absorption in the lymph compartment has been determined both qualitatively (Figure 3a: % of total fatty acids) and quantitatively (Figure 3b: mg/mL/g lipid intake).

The data showed that lymph was gradually enriched in n-3 PUFA over the 6 h following lipid administration, whatever the formulation.

From a qualitative point of view (Figure 3a), n-3 PUFA represents up to 25% of the total fatty acids when lipids are provided by the oil in bulk phase, and 32% when provided by the emulsion. From a quantitative point of view (Figure 3b), the lymphatic absorption of n-3 PUFA from native oil increased up to 6 h postprandial to reach a Cmax of 41 mg/mL/g lipid intake, while in emulsion form, the absorption kinetics of n-3 PUFA reached faster (Tmax = 4 h) a higher Cmax (91 mg/mL/g lipid intake).

The area under the curve (AUC) confirmed that the n-3 PUFA bioavailability was significantly (*p* < 0.05) higher when the algae was emulsified with GA (265 mg/mL/g of lipid intake.h), compared to the bulk phase (63 mg/mL/g of lipid intake.h).

EPA absorption

Figure 4 shows the absorption kinetics of EPA in the lymph compartment over the 6 h following lipid administration, both qualitatively (Figure 4a: % of total fatty acids) and quantitatively (Figure 4b: mg/mL/g lipid intake).

The data showed that lymph was gradually enriched in EPA over the 6 h following lipid administration.

From a qualitative point of view, EPA represented up to 0.85% or 1% when lipids were provided by the oil in bulk phase or emulsion with GA, respectively.

From a quantitative point of view, the lymphatic absorption of EPA from native oil increased up to 6 h postprandial to reach a Cmax of 1.3 mg/mL/g lipid intake, while when oil was emulsified with GA, the absorption kinetics of n-3 PUFA reached faster (Tmax = 4 h) a higher Cmax (2.8 mg/mL/g lipid intake).

The AUC confirmed that the EPA bioavailability was significantly (*p* < 0.05) higher with the emulsion form with GA (8.6 mg/mL/g of lipid intake.h), compared with the bulk phase form of the oil (2.5 mg/mL/g of lipid intake.h).

DHA absorption

Figure 5 shows the absorption kinetics of DHA in the lymphatic compartment over the 6 h following lipid administration, both qualitatively (Figure 5a: % of total fatty acids) and quantitatively (Figure 5b: mg/mL/g lipid intake).

The data showed that the lymph was gradually enriched in DHA over the 6 h following lipid administration. From a qualitative point of view, DHA represented up to 23.5 or 29.8% when lipids were provided by the oil in bulk phase or emulsified with GA, respectively.

From a quantitative point of view, the lymphatic absorption of DHA from native oil increased up to 6 h postprandial to reach a Cmax of 38.4 mg/mL/g lipid intake, while when the algae oil was emulsified with GA, the absorption kinetics of DHA reached faster (Tmax = 4 h) a higher Cmax (86.0 mg/mL/g lipid intake).

The AUC confirmed that the DHA bioavailability was significantly (*p* < 0.05) higher with the emulsion form (248 mg/mL/g of lipid intake.h), compared with the bulk phase form of the oil (56 mg/mL/g of lipid intake.h).

Absorption of the other main fatty acids from the oil

In addition to the interest of the n-3 PUFA contribution from the algae oil (emulsified or not), three other fatty acids characterized the oil, including the acids myristic (C14:0:9.1% of total fatty acids), palmitic (C16:0:21.4% of total fatty acids) and oleic (C18:1 n-9: 6% of total fatty acids).

The lymphatic enrichment of these fatty acids during the postprandial kinetics was possible only for myristic acid (0.7% of total fatty acids in lymph), as the two other fatty acids are not negligible endogenous fatty acids in lymph (palmitic: 27% and oleic: 7.7% of fatty acids in an alipidic lymph; internal data).

Figure 6 shows the absorption kinetics of myristic acid in the lymph compartment over the 6 h post-feeding, both qualitatively (Figure 6a: % of total fatty acids) and quantitatively (Figure 6b: mg/mL/g lipid intake).

The data showed that the lymph was gradually enriched in myristic acid over the 6 h following lipid administration.

From a qualitative point of view, myristic acid represented up to 6.9% or 5.1% of total fatty acids in lymph when lipids were provided by the oil in bulk phase or emulsified with GA, respectively. When provided in GA based emulsion (Figure 6a), myristic acid reaches a plateau later (T5h), compared to the bulk phase form (T3h).

From a quantitative point of view, the lymphatic absorption of myristic acid from native oil increased up to 6 h postprandial to reach a Cmax of 11.2 mg/mL/g lipid intake, while in emulsion form, the absorption kinetics of myristic acid reached faster (Tmax = 4 h) a higher Cmax (14.9 mg/mL/g lipid intake).

The AUC of the myristic acid bioavailability was more than two times higher (*p* = 0.1) when algae oil was emulsified with GA (43.4 mg/mL/g of lipid intake.h), compared to the oil in bulk phase (20.8 mg/mL/g of lipid intake.h).

## 4. Discussion

The objective of this study aimed to follow the impact of encapsulating n-3 PUFA from a microalgae oil with a natural emulsifier, gum acacia (GA), on the lymphatic bioavailability of n-3 PUFA. GA is a polysaccharide insoluble fiber, widely used for decades in food formulations to emulsify and stabilize lipid emulsions. GA has been also described to protect n-3 PUFA from oxidation, possibly due to its ability to coat the lipid droplets. Nevertheless, this specificity of GA to form a resistant crown-like shape around the lipid droplets could prove to be a brake on the lipolysis step, and therefore reduce the fatty acid absorption at the gut level. However, in vitro studies denoted a positive impact of GA on FA absorption. As far as we know, the use of GA as a surfactant has not yet been described in a complete scheme of digestion and absorption to support the previous in vitro investigations.

To answer this question, lipids were supplied to rats with a lymphatic duct fistulation in two physical forms of presentation, either in bulk phase or emulsified with GA. The FA absorption was followed during 6 h kinetics with a peculiar interest for DHA for its proven nutritional properties in human health. Lymph is the compartment of choice for studying the bioavailability of DHA since after digestion, lipids are directly absorbed through the enterocytes to enter the lymphatic way before being metabolized by the liver. The proportion of DHA is negligible in the lymph compartment since it represents less than 0.3% of total fatty acids in the interprandial period [17].

In this study, we clearly demonstrated that, regardless of its physical form of presentation, the consumption of algae oil significantly favored the fatty acid enrichment. The DHA reached a high level of enrichment in the lymph, i.e., 24 and 30% of total fatty acids, when lipids were provided in bulk phase and emulsion form, respectively. In addition, the lymphatic FA profile obtained after lipid administration denoted that the lymphatic enrichment in FA reflects the FA composition of dietary lipids, as previously reported elsewhere [13,57,58,59]. In order to increase the bioavailability of fatty acids, improving the lipolysis step is of importance and can be modulated by emulsified lipids. Indeed, physical characteristics and interfacial properties of the emulsion play an essential role in lipolysis. In this context, the role and the nature of the emulsifier used in lipid formulation is crucial. Recent works have highlighted the importance of the nature of the emulsifiers used in the emulsification process to improve or inhibit the lipolysis step and fatty acid absorption [12,17,60,61]. It is clear that lecithin-based emulsions favorably impact the digestion step at the root of an enhanced FA bioavailability, whereas casein or Tween 80 decreased FA bioavailability by reducing the lipolysis rate [13,18,62,63]. This could be mostly explained by a modification of the physical characteristic of the lipid droplets, such as the interfacial area and properties that are different according to the surfactant used during the emulsification process. In our study, algae oil was emulsified with GA, chosen for its capacity as an emulsifying agent and protector against the n-3 PUFA oxidation. GA, as a natural emulsifier, had not yet been studied in vivo and has been included in comparative studies but is particularly lacking.

In this study, we clearly demonstrated that the GA-based lipid-emulsion significantly increased in the lymph compartment, the AUC values for FA, and the DHA levels (+345%) compared to the bulk phase form. In fact, the AUC values were two times higher for total fatty acids and myristic acid and a 3.5 to 4 fold increase for longer chain n-3 FA, including EPA and DHA, respectively. Our results are consistent with previous works performed on animal [13,17,35,64] and human [14] models, demonstrating that lipid bioavailability was highly related to their lipid formulation [62,63]. Notably, various authors showed that the bioavailability of n-3 PUFA was two times higher when oils were provided in emulsion form compared to the bulk one [13,14,17,65]. These studies highlighted the interest of lecithin as an emulsifier to improve the bioavailability of alpha linolenic acid (ALA). In our study, the DHA-AUC was more favorable when emulsifying microalgae oil with GA (factor 4), compared to the AUC obtained for ALA when flaxseed oil was emulsified with lecithin (close to a factor of 2 [12,13,17]).

The increased AUC values were all the more true by the improvement of the Cmax values for both, total fatty acids and the overall n-3 PUFA. Compared to the oil in bulk phase, the emulsification of algae oil with GA improved 2.5 to 3 fold the Cmax, for both total fatty acids and the overall n-3 PUFA and by 25% the Cmax of myristic acid. These results are in agreement with the kinetic studies carried out on another n-3 PUFA, ALA, which also show an improvement in Cmax values (factor 1.5 with lecithin), but to a lesser extent compared to the Cmax DHA obtained in our study (factor 2.2 with GA). Our data revealed the interest for using GA to stabilize lipid formulations but also to improve the bioavailability of lipid of interests, such as n-3 LC PUFA.

Lipid digestion process is considered to be a limiting step in the bioavailability of fatty acids. GA has the particularity of protective coating all around the lipid droplets, which gives it an antioxidant property, particularly with regard to n-3 LC-PUFA. Studies based on pickering emulsions have showed that an anti-oxidant particle was more effective when it was located at the lipid interface rather than located within the lipid droplet [47,66]. However, this specific structure, by forming a coat around the lipid structure, also provides peculiar resistance to acidic pH, as observed during the gastric phase. Indeed, some hydrocolloids are resistant to enzymatic degradation in the stomach and small intestine, which allow them to retain their polymeric form throughout the stomach and small intestine. In this context, GA could have a putative inhibitory effect on the lipolysis step, and thus ultimately reduce the absorption of the fatty acids of interest provided by the microalgae oil. It has been reported that physicochemical factors of dietary fiber can affect nutrient absorption. For example, based on recent data, dietary fiber located in the upper gastrointestinal tract would decrease the rate of intestinal absorption of nutrients and can be detrimental due to the reduction absorption of essential micronutrients [67,68].

In light of our lymphatic results, marked by a significant enrichment in lipid and DHA, this crown-like shape, formed by GA, does not seem to have any impact on the overall lipolysis step or to reduce the absorption of fatty acids. Herein, the improved bioavailability was linked to the broader interface generated by emulsification, which plays a crucial role in the catalysis reaction. As an interfacial enzyme, pancreatic lipase needs a certain oil/water interface area to operate [23,60,69,70], where the number of the lipid droplets and also their diameter take part. The surface is important for the physical stabilization of emulsions and is linked to the properties of the interfacial layer of the emulsion. In our study, by using GA as an emulsifier, we provided an emulsion with 3µm maximum lipid droplet size. It has been reported in vitro and in vitro studies that fine emulsions with “small” lipid droplets (diameter < 1 µm) were hydrolyzed faster by pancreatic lipase compared to coarse ones (>20 µm) [24,31,71]. The micrometer size of the GA-emulsion improved the lipid interface and makes it available to favor the accessibility of lipase onto the lipid droplets, accelerating the lipolysis of n-3 LC. Thus, the greater the surface area, the greater and faster the TG hydrolysis [25,72,73,74]. The crown-like shape obtained with GA seems to preserve the micronic lipid structure during the lipolysis step, which is essential for pancreatic lipase without hindering its accessibility at the lipid interface. GA is one of the most favorable candidates in emulsifiers for improving the granulometry and stability of PUFA formulations by limiting coalescence, particularly during pH variations, as observed in vivo during gastrointestinal digestion [29,52].

On another hand, we determined the hourly fatty acid uptake and the peak of FA absorption with regard to the kinetics of lipid absorption in lymph. By considering the shape of the kinetic curve, we observed that the Tmax obtained at 6 h postprandial was later compared to the Tmax previously observed for vegetable oils (4 h postprandial) [13,17,18]. This delay could be explained by the steric hindrance steric induced by n-3 LC PUFA, which limits the enzymatic access [23,57] and slows down the in vitro lipolysis [69,75]. Indeed, the activity of digestive lipases, as pancreatic lipase, mainly depends on the composition and interfacial properties, but also on the FA nature [15], which conditions the lipolysis level. According to some studies, medium chain FA are privileged substrates for lipases, whereas n-3 LC-PUFA interferes with the access of the enzyme on the lipid interface, resulting in a dramatically impaired lipolysis rate, and thus FA intestinal-absorption [58,69,76]. In our study, as a matter of fact, the lymphatic absorption of dietary DHA showed that during the lipid digestion process, pancreatic lipase was not inhibited by the steric conformation of DHA from algae oil but induced a reduced or delayed catalytic reaction, compared to shorter or less unsaturated FA chains in vegetable oils [69,75].

However, when algae oil was emulsified with GA, the peak of FA absorption was earlier for the emulsion (Tmax = 4 h postprandial) than that observed for bulk oil (Tmax = 6 h postprandial) and similar to that previously observed for emulsified vegetable oil (Tmax = 4 h postprandial) [13]. These studies highlighted that the process of lipid emulsification might advance the peak of lymphatic absorption of shorter lipid chains both in rats [13,17] and in humans [14,64]. They also reported that the emulsification process improved not only the ALA bioavailability (by a factor 2) but also the rate of appearance in the lymph, marked by an earlier Tmax of 1 h.

GA has demonstrated a benefit regarding the bioavailability of nutritional lipids. This benefit requires further studies in humans for confirmation and investigation in chronic approaches.

## 5. Conclusions

To conclude, our data clearly demonstrated that lipid emulsification with GA ameliorated quantitatively and qualitatively the lymph status in n-3 PUFA by a factor of 4.2, and more significantly than with soy lecithin, which was the emulsifier that best improved lipid bioavailability in the comparative studies [12]. The lymph quantity of other triglycerides as myristic acid was also improved by a factor of 2.1. Lipidic formulation with GA and the achievement of a micrometric droplet mesh size represents a promising interest in improving its bioavailability.

## Figures and Tables

**Figure 1 biomolecules-12-00975-f001:**
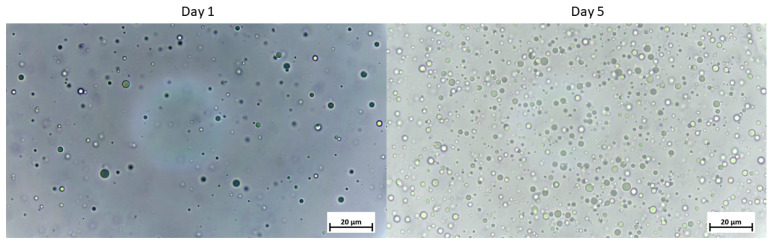
Five-day monitoring of lipid droplet size by optical microscopy (obj.× 100) of the oil emulsion with gum acacia, with homogenization.

**Figure 2 biomolecules-12-00975-f002:**
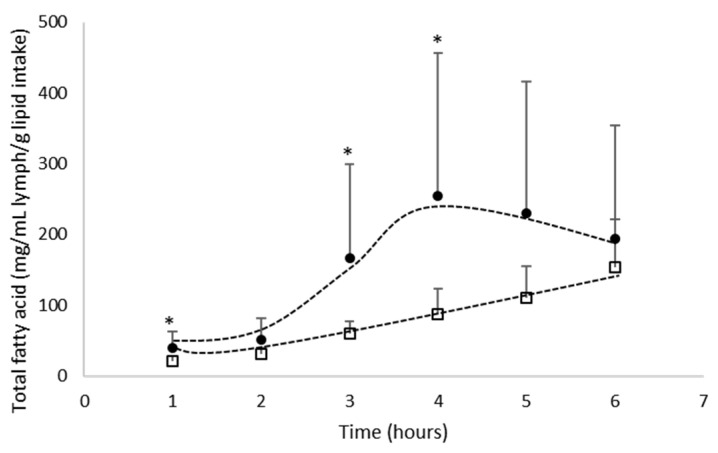
Kinetics of intestinal absorption of fatty acids (mg/mL lymph/g of lipid intake) in rats submitted to a lymphatic duct’s fistula (*n* = 8 rats/group) and orally supplied with 200 mg of lipids from an oil either in bulk phase (□) or emulsified with gum acacia (●) over a 6 h period. Data are presented as their means ± standard deviation *. For the same time period, data are significantly different between the two groups (*p* < 0.05; Student’s *t*-test).

**Figure 3 biomolecules-12-00975-f003:**
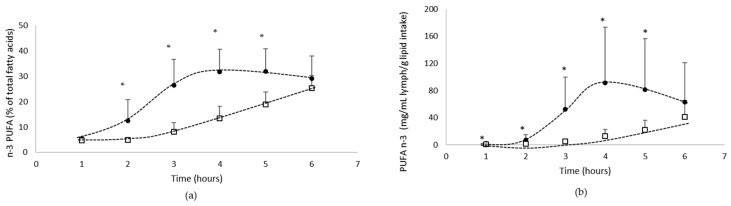
Kinetics of intestinal absorption of n-3 PUFA ((**a**): % of total fatty acids and (**b**): mg/mL lymph/g of lipid intake), in rats submitted to a lymphatic duct’s fistula (*n* = 8 rats/group) and orally supplied with 200 mg of lipids from an oil either in bulk phase (□) or emulsified with gum acacia (●) over a 6 h period. Data are presented as their means ± standard deviation *. For the same time period, data are significantly different between the two groups (*p* < 0.05; Student’s *t*-test).

**Figure 4 biomolecules-12-00975-f004:**
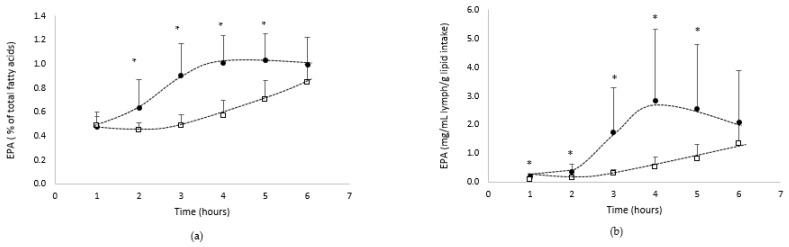
Kinetics of intestinal absorption of EPA ((**a**): % of total fatty acids and (**b**): mg/mL lymph/g of lipid intake), in rats submitted to a lymphatic duct’s fistula (*n* = 8 rats/group) and orally supplied with 200 mg of lipids from an oil either in bulk phase (□) or emulsified with gum acacia (●) over a 6 h period. Data are presented as their means ± standard deviation *. For the same time period, data are significantly different between the two groups (*p* < 0.05; Student’s *t*-test).

**Figure 5 biomolecules-12-00975-f005:**
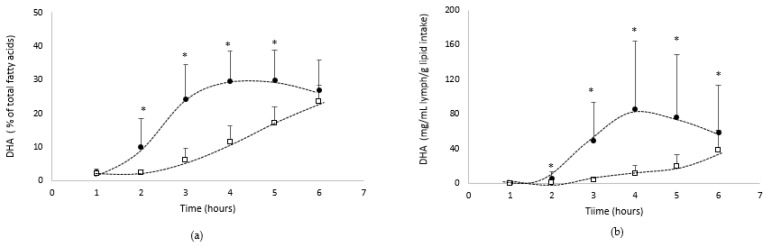
Kinetics of intestinal absorption of DHA ((**a**): % of total fatty acids and (**b**): mg/mL lymph/g of lipid intake), in rats submitted to a lymphatic duct’s fistula (*n* = 8 rats/group) and orally supplied with 200 mg of lipids from an oil either in bulk phase (□) or emulsified with gum acacia (●) over a 6 h period. Data are presented as their means ± standard deviation *. For the same time period, data are significantly different between the two groups (*p* < 0.05; Student’s *t*-test).

**Figure 6 biomolecules-12-00975-f006:**
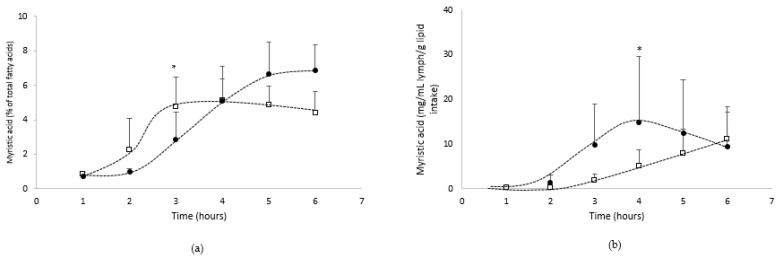
Kinetics of intestinal absorption of myristic acid ((**a**): % of total fatty acids and (**b**): mg/mL lymph/g of lipid intake), in rats submitted to a lymphatic duct’s fistula (*n* = 8 rats/group) and orally supplied with 200 mg of lipids from an oil either in bulk phase (□) or emulsified with gum acacia (●) over a 6 h period. Data are presented as their means ± standard deviation *. For the same time period, data are significantly different between the two groups (*p* < 0.05; Student’s *t*-test).

**Table 1 biomolecules-12-00975-t001:** Fatty acid profile of the oil in bulk phase and emulsified with gum acacia.

	Oil	Emulsion
Total fatty acid (%)		
SFA *	30.9	33.4
12:0	0.7	0.8
14:0	8.6	9.1
16:0	20.2	21.4
18:0	0.7	0.7
MUFA *	9.8	10.5
14:1	0.1	0.1
16:1	4.0	4.3
18:1	5.8	6.0
PUFA *	59.0	55.5
PUFA n-6 *	9.7	9.4
18:2(n-6)	1.0	1.1
20:4(n-6)	0.2	0.2
22:5(n-6)	8.2	7.9
PUFA n-3 *	49.3	46.1
18:4(n-3)	0.2	0.3
20:5(n-3)	1.5	1.5
22:5(n-3)	0.1	0.3
22:6(n-3)	47.3	43.9

* SFA: saturated fatty acid, MUFA: monounsaturated fatty acid, PUFA: polyunsaturated fatty acid from n-6 and n-3 series.

**Table 2 biomolecules-12-00975-t002:** Glyceridic composition of the microalgae oil.

Parameters	
	% of Total Lipids
Polymer	0.2
Triglyceride	97.2
Diglyceride	<0.1
Monoglyceride	0.9
Free fatty acids and others (sterol, FAME *)	1.6

* FAME: fatty acid methyl esters.

**Table 3 biomolecules-12-00975-t003:** Peroxyde and P-anisidine values.

Parameters	Oil	Emulsion
Peroxyde value (mEq O_2_/kg)	1.5 ± 1.0	5.3 ± 2.1
P-anisidine value	5.1	7.1

**Table 4 biomolecules-12-00975-t004:** Summary data for the maximal concentration (Cmax; mg/mL), area under the curve (AUC; mg/mL.h) and Tmax (h) for lymphatic main fatty acids in animals submitted to unemulsified versus emulsified oil.

		Oil	GA-Emulsion	
Total FA *	AUC *	375.1	820.9	^$^
	Cmax *	153.6	254.9	^£^
	Tmax *	6 h	4 h	^¤^
n-3 PUFA *	AUC	62.9	265.2	^$^
	Cmax	41	91.1	^£^
	Tmax	6 h	4 h	^¤^
EPA *	AUC	2.5	8.6	^$^
	Cmax	1.3	2.8	^£^
	Tmax	6 h	4 h	^¤^
DHA *	AUC	55.8	248.2	^$^
	Cmax	38.4	86	^£^
	Tmax	6 h	4 h	^¤^
Myristic acid	AUC	20.8	43.4	^$^
	Cmax	11.2	14.9	^£^
	Tmax	6 h	4 h	^¤^

FA: fatty acid, PUFA: polyunsaturated fatty acid from n-6 and n-3 series, EPA: eicosapentaenoic acid, DHA: docosahexaenoic acid. AUC: area under the curve (mg/mL.h), Cmax: maximal concentration (mg/mL) and Tmax maximal time to reach the Cmax value (h). Rats received a unique bolus of the oil in the following two forms: either in bulk phase or emulsion; lymph was collected hourly for 6 h postprandial. Data are represented by their mean ± standard deviation SD (*n* = 8 rats/group). Between the two forms of intake (oil or emulsion), means marked with a different superscript (£,¤,$) are significantly different (* *p* < 0.05, Student’s *t*-test).

## Data Availability

Data availability statements are available under ITERG and Nexira contentment.

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
