# Peer review of "Effect of Gum Acacia on the Intestinal Bioavailability of n-3 Polyunsaturated Fatty Acids in Rats"

_biomolecules, 2022, doi:10.3390/biom12070975_

Round 1

Reviewer 1 Report

Authors investigated the influence of gum acacia as emulsifier on the intestinal absorption of n-3 poly 2 unsaturated fatty acids. Below, I suggest a number of changes to this manuscript: the writing, as well as scientific content and experimental design need to be thoroughly revised. 

Please comment of the role of intestinal lymphatic transport and its role in lipid absorption from the gut in the introduction. 

Extensive edits to the grammar, language, spacing (between words, number and units, references), punctuation and appropriate abbreviations are necessary (for instance, abbreviation of PUFA, EPA, DHA etc.). 

This sentence need to be rewritten: " But recently, the process of lipid emulsification has been described to improve the intestinal 38 uptake of oils and lipophilic[11–15] and more, the bioavailability of n-3 PUFAs whether 39 in lymph or in plasma[12,16–22]."

gum acacia (GA)-whether it is capital A or not, it should be consistent throughout the manuscript and title. 

"Hydrocolloids are water-soluble polymers 59 with a wide variety." - wide variety of what?

"Mw; from 0.02 to 11 x106 g.mol−1" is it 10 to the power of 6 or 106?? Also multiplication sign should be introduced  instead of a dot sign. 

Please change "orally submitted" to "orally administered DHA..." in the abstract. 

There should either be/not be a space between the word and reference (please keep it consistent in accordance to journal style). 

8-week old male Wistar rats weight ~250g or less, yet in the methods it is declared that they were 300-350g. What is the reason for this?

What is the reason for not fasting the rats overnight prior to procedure?

In figure 2, the SD of group receiving gum acacia is very high, what might be the reason for this?

Figure 6 des not contain labels A and B. Please explain behavior of myristic acid in Figure 6a.  

Were EPA and DHA administered orally (I believe they were, because you are following the effect on intestinal permeability), and if so please describe the administration procedure?

Please describe briefly the surgical procedure in the methods section. 

Postprandial is usually one word. 

Description of statistical tests used, statistical software etc. should be provided. 

The importance of this study and it's contribution in the clinical sense should be discussed, given the vast knowledge of the role of gum acacia on lipid permeability. What distinguishes your study from others? 

Author Response

We thank Reviewer 1 for taking the time to review the article and her/his report and constructive comments. As requested, the changes have been done in the revised version of the manuscript. Please find enclose our responses to the questions asked.

Reviewer 2 Report

In pharmacology, bioavailability is the fraction of an administered drug that reaches the systemic circulation. This is normally given as a percentage. Thus, the use of the term “bioavailability” in this manuscript is incorrect as the concentrations of DHA and EPA are presented but not their bioavailability. 

Line 31: EPA is eicosapentaenoic acid not eicosanic acid.

Introduction – the authors should mention that EPA/DHA are usually in form of triglycerides (97.2% in Table 2) so that lipolysis makes sense. 

At end of Introduction, please add hypothesis.

Lines 108-9: Please include microalgal source (Schizochytrium?) and processing details. This reference suggests that the microalgal oil is from Schizochytrium https://cdn.tradingfoe.com/attachment/file/7105/OMEGAVIE_DHA_400_ALGAE_QS-L_TDS.PDF but this needs to be elaborated in the text. What does “QualitySilver” mean and what is the importance of this term?

Lines 185-187: Were fatty acids administered by oral gavage or by injection? Please give details of method of administration.

How were EPA/DHA doses chosen? Are they relevant to the doses recommended for humans, using Reagan-Shaw calculations?

Line 344: Do the authors mean “encapsulating” or “microencapsulating”?

Author Response

We thank Reviewer  2 for taking the time to review the article and her/his report and constructive comments. As requested, the changes have been done in the revised version of the manuscript. Please find enclosed our responses to the questions asked.

Reviewer 3 Report

The manuscript by Leslie Couëdelo et al. investigated the effect of gum acacia on the intestinal bioavailability of n-3 polyunsaturated fatty acids in rats. The study is of interest to some of the readers. I have the following suggestions and comments:

1, The authors must further revise the tables. All tables must be in a three-line format. 

2, Some of the figures should be combined. For example, figure 2. 

3, As for the statistics, the authors must double-check. Is it suitable to use t-test student? The authors should explain. 

4, How could polysaccharides such as gum acacia affect the bioavailability of n-3 polyunsaturated fatty acids? The authors should further explain the mechanisms and if it is possible, the authors should add another figure to illustrate this point. 

Author Response

We thank Reviewer 3 for taking the time to review the article and her/his report and constructive comments. As requested, the changes have been done in the revised version of the manuscript. Please find enclosed our responses to the questions asked.

Round 2

Reviewer 1 Report

Thank you for addressing my comments and concerns. I have a couple of further suggestions for authors: 

Figure 2 demonstrates kinetics of intestinal absorption of fatty acids following oral administration, as revealed by the sampling of the lymph from the lymphatic duct’s fistula? Am I correct? If so this should be written clearly.  Also the authors state that there is significant difference, however the standard deviation is so high, I wonder how statistically different it is. Please elaborate a little bit about the meaning of this difference. 

What is GA in line 30?

line 33 should state "of developing"

Line 57-62 needs some work in order to improve clarity and flow. Line space between "withinenterocyte"

Line 287 needs to be amended for clarity. 

Author Response

Dear Reviewer 1,

Please find enclosed the response to your comments.

Thank you for reviewing our manuscript.

Best regards

Reviewer 2 Report

I don't agree with the authors on the definition of bioavailability but I accept the they have made their interpretation clear in the manuscript. 

One small question: what does "gangue" mean to the authors? The online dictionary defines gangue as the waste product in metal ore extraction, which doesn't seem appropriate to this study. 

Author Response

Dear Reviewer 2,

Please find enclosed the response to your comments.

Thank you for reviewing our manuscript.

Best regards

Reviewer 3 Report

The authors have revised the manuscript, and there is no other comments.

Author Response

Dear Reviewer 3,

Please find enclosed the response to your comments.

Thank you for reviewing our manuscript.

Best regards
